# miR-499a rs3746444 A>G Polymorphism Is Correlated with Type 2 Diabetes Mellitus and Diabetic Polyneuropathy in a Romanian Cohort: A Preliminary Study

**DOI:** 10.3390/genes14081543

**Published:** 2023-07-27

**Authors:** Emilia Burada, Maria-Magdalena Roșu, Raluca Elena Sandu, Florin Burada, Mihai Gabriel Cucu, Ioana Streață, Bianca Petre-Mandache, Gabriela Popescu-Hobeanu, Monica-Laura Cara, Anca-Maria Țucă, Elena Pinoșanu, Carmen Valeria Albu

**Affiliations:** 1Department of Physiology, University of Medicine and Pharmacy of Craiova, 200349 Craiova, Romania; emilia.burada@umfcv.ro (E.B.); ancabirau94@yahoo.com (A.-M.Ț.); 2Department of Neurology, Clinical Hospital of Neuropsychiatry Craiova, 200473 Craiova, Romania; raluca.sandu@umfcv.ro (R.E.S.); elenapinosanu@yahoo.com (E.P.); carmen.albu@umfcv.ro (C.V.A.); 3Department of Diabetes, Nutrition and Metabolic Diseases, Emergency Clinical County Hospital Craiova, 200642 Craiova, Romania; magda.sandu23@yahoo.com; 4Department of Biochemistry, University of Medicine and Pharmacy of Craiova, 200349 Craiova, Romania; 5Laboratory of Human Genomics, University of Medicine and Pharmacy of Craiova, 200638 Craiova, Romania; mihai.cucu@umfcv.ro (M.G.C.); ioana.streata@umfcv.ro (I.S.); aknaib86@gmail.com (B.P.-M.); gmph94@gmail.com (G.P.-H.); 6Regional Centre of Medical Genetics Dolj, Emergency Clinical County Hospital Craiova, 200642 Craiova, Romania; 7Doctoral School, University of Medicine and Pharmacy of Craiova, 200349 Craiova, Romania; 8Department of Public Health, University of Medicine and Pharmacy of Craiova, 200349 Craiova, Romania; monica.cara@umfcv.ro; 9Department of Neurology, University of Medicine and Pharmacy of Craiova, 200349 Craiova, Romania

**Keywords:** type 2 diabetes mellitus, diabetic polyneuropathy, microRNAs, single nucleotide polymorphism, genotype

## Abstract

Type 2 diabetes mellitus (T2DM) is a common metabolic disorder that results from complex interactions of both environmental and genetic factors. Many single nucleotide polymorphisms (SNPs), including noncoding RNA genes, have been investigated for their association with susceptibility to T2DM and its complications, with little evidence available regarding Caucasians. The aim of the present study was to establish whether four miRNA SNPs (miR-27a rs895819 T>C, miR-146a rs2910164 G>C, miR-196a2 rs11614913 C>T, and miR-499a rs3746444 A>G) are correlated with susceptibility to T2DM and/or diabetic polyneuropathy (DPN) in a Romanian population. A total of 167 adult T2DM patients and 324 age- and sex-matched healthy controls were included in our study. miRNA SNPs were detected by real-time PCR using a TaqMan genotyping assay. A significant association with T2DM was observed only for the miR-499a rs3746444 A>G SNP in all the tested models, and the frequencies of both the miR-499a rs3746444 AG and the GG genotypes were higher in the T2DM patients compared to the controls. No correlation was observed for the miR-27a rs895819 T>C, miR-146a rs2910164 G>C, or miR-196a2 rs11614913 C>T SNPs in any genetic model. When we assessed the association of these SNPs with DPN separately, we found a positive association for the miR-499a rs3746444 SNP in both codominant and dominant models (OR 6.47, 95% CI: 1.71–24.47; OR 2.30, 95% CI: 1.23–4.29, respectively). In conclusion, this study shows that miR-499a rs3746444 A>G may influence both T2DM and DPN susceptibility, with carriers of the GG genotype and the G allele being at an increased risk in the Romanian population.

## 1. Introduction

Diabetes mellitus is a metabolic disorder which is well-recognized and accepted as a global health issue due to its widespread, rapidly growing prevalence and associated morbidity and mortality [1,2]. Those who have diabetes mellitus are at a higher risk of developing microvascular and macrovascular complications (e.g., nephropathy, retinopathy, neuropathy, diabetic foot, hypertension, dyslipidemia) which contribute to the devastating human, social, and economic impact of this disease [3,4].

Type 2 diabetes mellitus (T2DM) is the most common type, accounting for 85–95% of all diabetes cases, and is characterized by progressively higher glycemia, caused by both insulin resistance and β-cell dysfunction [5,6]. T2DM is a multifactorial and heterogeneous disease that results from complex interactions of both environmental and genetic factors. The underlying mechanism of T2DM remains incompletely understood, although some risk factors (e.g., age, ethnicity, family background, high body mass index, insufficient physical activity, and smoking) or more susceptibility genetic loci (e.g., TCF7L2, CDKN2A/B PPARG, KCNJ11, KCNQ1, and IGF2BP2) have been identified [7,8].

Many genetic variants have been investigated for their association with susceptibility to diabetes and its complications [9], including noncoding RNA genes [10]. microRNAs (miRNAs) are small RNAs, consisting of 20 to 25 nucleotides, that affect cellular functions via modulating transcription, translation efficacy, and/or degradation of specific messenger RNAs (mRNAs), targeting sequences in their 3′ untranslated region and thus changing protein levels [11,12]. miRNA molecules have been involved in the pathogenesis of different diseases including T2DM [13]; it is estimated that miRNA negatively regulates the expression of more than 60% of protein-coding genes [14]. There is increasing evidence that miRNAs influence many aspects of insulin synthesis, glucose metabolism, and homeostasis, mostly by affecting pancreatic islet β-cell biological processes, including cell development and proper function [10]. Single nucleotide polymorphisms (SNPs) located within miRNA genes may affect the expression of target genes or the downstream action of the corresponding miRNAs [12] and thus interfere with susceptibility to T2DM and its complications [13]. Previous studies regarding the association between miRNA variants and T2DM were focused on Asian populations and produced conflicting results [10]. To date, little evidence is available on Caucasians, and to our knowledge no studies have investigated the association between miRNA SNPs and susceptibility to T2DM in an Eastern European country.

To address this issue, the aim of this preliminary study was to establish whether miR-27a rs895819 T>C, miR-146a rs2910164 G>C, miR-196a2 rs11614913 C>T, and miR-499a rs3746444 A>G SNPs are correlated with susceptibility to T2DM in a Romanian population. Secondly, we investigated the existence of any association between alleles or genotypes of the identified SNPs with diabetic polyneuropathy (DPN).

## 2. Materials and Methods

### 2.1. Patients

In this case-control study, a total of 167 adult T2DM patients and 324 healthy controls were included. The T2DM subjects were recruited among patients admitted to the Department of Neurology of the Hospital of Neuropsychiatry of Craiova, Romania. The diagnosis of T2DM was made on the basis of criteria established by the American Diabetes Association [15]. Patients with any type of cancer, autoimmune diseases, nephropathy, or severe liver disease history were excluded. 

Medical data, including age, sex, body mass index (BMI), blood glucose at the moment of blood sample collection, glycosylated hemoglobin (HbA1c), or presence of complications was collected for T2DM subjects. The T2DM diagnosis was based on fasting plasma glucose (FPG) levels ≥ 7.0 mmol/L (126 mg/dL) or a 2-h postglucose level ≥ 11.1 mmol/L (200 mg/dL) by either a random elevated FPG value on more than one occasion, an oral glucose tolerance test, or HbA1c ≥ 6.5% (48 mmol/mol). The BMI was calculated on the basis of height and weight and using the formula BMI = weight (kg)/height^2^ (m^2^). According to these criteria, the subjects were divided into three groups: underweight (<18.5 kg/m^2^), normal weight (18.5–25 kg/m^2^), and overweight and obese (>25 kg/m^2^). DPN diagnosis relied on the presence of at least two abnormal findings among neuropathic symptoms, or thresholds for vibration and thermal perception. In a subgroup of 40 patients with DPN, nerve conduction studies were performed on the median, ulnar, tibial, and peroneal motor nerves bilaterally and on the sural, median, and ulnar sensory nerves bilaterally using a Neuropack S1 MEB-9400 EMG system (Nihon Kohden, Tokyo, Japan).

The control group consisted of 324 ethnically and age- and sex-matched unrelated healthy subjects, with no family history of diabetes mellitus nor any type of cancer or chronic inflammatory disorders. 

The study complies with the Declaration of Helsinki and was performed with the approval of the Ethics Committee of the University of Medicine and Pharmacy of Craiova, Romania (no. 113/03.06.2022). Each participant signed a written consent form before inclusion in the study.

### 2.2. miRNA SNP Genotyping

Whole-blood samples from all patients were collected in EDTA (ethylenediaminetetraacetic acid) tubes and stored at 4 °C for no more than 5 days before being processed. A Wizard^®^ Genomic DNA Purification Kit (Promega, Madison, WI, USA) was used to extract genomic DNA from peripheral lymphocytes following the manufacturer’s instructions. The DNA was later assessed for quality and yield using the NanoDrop-2000 Micro Volume UV Spectrophotometer (NanoDrop Corporation, Waltham, MA, USA). Four miRNA SNPs: (miR-27a rs895819 T>C (assay C_3056952_20), miR-146a rs2910164 G>C (assay C_15946974_10), miR-196a2 rs11614913 C>T (assay C_31185852_10), and miR-499a rs3746444 A>G (assay C_2142612_40) were detected by RT-PCR using predesignated TaqMan probes (Applied Biosystems, Foster City, CA, USA). The amplification and detection were performed using a ViiA7 real-time PCR system (Thermo Fischer Scientific, Waltham, MA, USA), following standard recommended protocols: denaturation of the dsDNA template for 10 min at 95 °C, followed by annealing of primers and extension of dsDNA molecules with allele detection in 50 cycles of 95 °C for 15 s and 62 °C for 1 min. Ten percent of the samples were genotyped and analyzed twice in order to assess the quality of the RT-PCR reactions. The concordance rate was 100%, with each run containing positive and negative controls.

### 2.3. Statistical Analysis

Descriptive statistics are used to describe the basic features of our study data. Continuous variables are presented as the mean ± SD, and categorical data is reported as values or percentages. The Hardy–Weinberg equilibrium (HWE) was verified using the Chi-squared test. The allele and genotype frequencies of the SNPs were calculated by direct counting. The associations of each individual SNP with T2DM or DPN were calculated as odds ratio (OR) with 95% confidence interval (CI) using logistic regression. miRNA SNPs were compared between the groups under codominant, recessive, dominant, and allelic inheritance models. The ANOVA test was also used to compare the differences in metabolic factors between SNP genotypes in the T2DM group. A *p* value of less than 0.05 was considered statistically significant for all the tests. The data was analyzed using SPSS Statistics for Windows, Version 22.0 (IBM SPSS Statistics for Windows, Version 22.0. Armonk, NY, USA: IBM Corp).

## 3. Results

We genotyped four miRNA SNPs in a total of 491 subjects, including 167 T2DM patients and 324 healthy controls. The groups were matched for age and sex, with no statistically significant differences between the T2DM patients and the controls (*p* > 0.05). The parameters recorded for the T2DM subjects are presented in Table 1. The median BMI was 29.71 (21.8–40 kg/m^2^), and FPG and HBA1c had median levels of 172 mg/dL and 7.2% respectively. Eighty-nine T2DM patients developed DPN.

For each SNP, the genotype distributions were consistent with those predicted by the Hardy–Weinberg equilibrium in the controls, signifying no genetic drift or selective advantage (*p* > 0.05) (Table 2).

### 3.1. miRNA SNPs and T2DM

The minor allele of each SNP was considered the risk variant, opposed to the wild-type allele. Genotypes, allele frequencies, and their association with T2DM risk are shown in Table 3. A significant association was observed only for the miR-499a rs3746444 A>G SNP in all the tested models. A higher frequency of both miR-499a rs3746444 AG and GG genotypes was found in T2DM patients compared to controls. The strongest association was found in the codominant model, with the relative risk for carriers of the GG genotype being 3.34 (OR 3.34, 95% CI: 1.54–7.28) compared to the more frequent AA genotype. Our results showed no correlation between the T2DM cases and the controls for the remaining SNPs in any studied genetic model (codominant, dominant, recessive, or allelic). 

### 3.2. miRNA SNPs and DPN

We assessed the association of these SNPs with the DPN separately (Table 4) and found a positive association for the miR-499a rs3746444. In the codominant and dominant models, the frequency of the miR-499a rs3746444 GG genotype and the carriers of the G allele (AG + GG genotype) was significantly higher in the T2DM patients with DPN compared to the T2DM patients without DPN (OR 6.47, 95% CI: 1.71–24.47; *p* = 0.002, and OR 2.30, 95% CI: 1.23–4.29; *p* = 0.008 respectively). We found no significant differences for the remaining miRNA SNPs.

### 3.3. miRNA SNPs and Metabolic Factors

No significant differences were detected in the patient group between different rs895819, rs11614913, and rs3746444 miRNA SNPs and age, BMI, glycemia, and HbA1c (Table 5). For miR-146a rs2910164 SNP, there was a statistically significant difference between groups as determined by one-way ANOVA (F = 3.315, *p* = 0.039). A Tukey post hoc test revealed that the BMI was statistically significantly lower for the CC genotype (25.85 ± 3.25, *p* = 0.043) compared to the GG genotype (30.19 ± 4.47). There was no statistically significant difference between the GC and GG (*p* = 0.436) or CC (*p* = 0.136) groups. Also, no significant differences were detected between different rs2910164 genotypes and age, glycemia, and HbA1c in the patient group.

## 4. Discussion

In this association case-control study, we assessed the correlation between four miRNA SNPs with T2DM and DPN in a Romanian population. The selection of these SNPs was based on their involvement in common multifactorial diseases, including T2DM [10,16], and on previous findings that showed different expression patterns.

Many case-control studies have investigated the link between miRNA SNPs and susceptibility to T2DM, with some studies reporting associations between the two and other studies showing no significant effect.

We found a positive association only in the case of the miR-499a rs3746444 SNP, with the carriers of the GG genotype and the G allele being associated with an increased susceptibility to both T2DM and DPN. This SNP consists of an A to G nucleotide substitution (A>G) which generates a mismatch in the stem loop sequence of the miR-499 precursor. It has been hypothesized that this could influence the maturation of both miR-499a-5p and miR-499a-3p, the binding of miR-499a-3p to its targets, and miR-499a expression [17,18].

miR-499a influences both the PI3K/AKT/GSK signaling pathway and glycogen synthesis by targeting PTEN [19] and the mitochondrial dynamics attenuating Drp1 activity through its target calcineurin [20]. 

In a study by Ciccacci et al., the expression of miR-499a seems to be downregulated in patients with DPN, but no association was found between rs3746444 and miR-499a expression in patients with T2DM and DPN compared to patients who are negative for these neurological complications [21]. miR-499a rs3746444 might impair mitochondrial biogenesis in T2DM patients by reducing the mtDNA copy number, mainly when DPN is present [22,23]. Furthermore, in a cohort of Italian people with T2DM which included 150 subjects, the GG genotype was associated with an increased risk to both DPN and cardiovascular autonomic neuropathy [24]. This SNP was associated with T2DM in a Korean cohort only when combined with miR-27a [25], whereas no correlation was found in any model or combinations in a Chinese study [26]. 

We did not find any association between miR-27a rs895819, miR-146a rs2910164, or miR-196a2 rs11614913 and T2DM or DPN in our cohort.

The replacement of the wild-type A allele with G (or T with C on the opposite strand) results in mir-27a rs895819 SNP and contributes to an aberrant process from pri-miR-27a to pre-miR-27 [27] affecting the amount of miR-27a [28]; a higher miR-27a expression was observed for AG and GG genotype carriers compared to the wild-type genotype. 

miR-27a was found to be highly expressed in adipose tissue, and by targeting the peroxisome proliferator-activated receptor γ (PPARγ) gene and reducing its expression it negatively regulates adipogenesis [29]. Also, in adipose tissue of a rat model of type 2 diabetes, the miR-27a has been found to be upregulated in high glycemic conditions [30]. Moreover, an altered miR-27a expression was detected in patients diagnosed with metabolic syndrome and T2DM, showing a strong positive correlation with fasting glucose levels [31].

A dose–allele effect was proposed, with the carriers of the G allele and the GG genotype showing a protective effect against T2DM in an Italian T2DM cohort [32]. Surprisingly, the same group reported a higher risk of early development of cardiovascular autonomic neuropathy for the same G allele, suggesting different pathways in T2DM and diabetic cardiovascular autonomic neuropathy [33]. A protective role for rs895819 SNP against T2DM was reported in Iranian [34] and Korean [25] cohorts, respectively. In contrast, no correlations were detected in different models for Chinese populations [35,36]. However, the stratified analysis of Wang et al. revealed that the CC genotype of miR-27a rs895819 was significantly associated with an increased risk of T2DM in the obese and younger groups of a Chinese Han population [36]. Conflicting results were also observed in two meta-analyses. In the first meta-analysis, the CC genotype of miR-27a rs895819 was associated with a significantly decreased risk of T2DM in Asian populations compared with the TT genotype [37]; the second meta-analysis showed no significant result in the overall population, but subgroup analysis revealed that minor allele significantly decreased the risk of T2DM in Caucasians [38].

The miR-146a rs2910164 G>C SNP causes an unstable structure of pre-miR-146a [39], which leads to a decrease in both pre/mature miR-146a, with the presence of C allele being associated with a low level of miR-146a expression [40]. The reduced level of miR-124 could promote progression to T2DM through interactions with NF-κB, TRAF-6, and cytokines (TNFα and IL-6), which lead to insulin resistance, poor glycemic control, and a pro-inflammatory state [41,42]. A decreased expression of miR-146a has been reported in T2DM patients compared to healthy controls [43,44] and was correlated with the presence of rs2910164 variant [39]. miR-146a rs2910164 SNP was associated with T2DM and its cardiovascular risk factors in an Iranian population [45], while Ciccacci et al. failed to detect any association with T2DM in an Italian population [32]. Furthermore, a protective effect of this variant for T2DM was also reported [26]. The findings are controversial even in the same population, with rs2910164 being associated [35] or not [36,46] with T2DM in Chinese Han people. The minor allele of rs2910164 was correlated with the risk of T2DM in the overall population [38], in Caucasians [46], and in Latin Americans [37]. Another meta-analysis failed to detect a significant association between miR-146a SNP rs2910164 and T2DM in the overall population, in Asians, or in Caucasians under multiple genetic models [47].

The rs11614913 T>C variant in the miR-196a2 gene is located at the 3p regions in mature miR [48] and influences the processing of miR-196a2 and its function [49]; it is speculated that the rs11614913 C allele also increases the expression of miR-196a2 [49,50]. miR-196a is involved in T2DM pathogenesis through activation of the AKT signaling pathway [51]. A case-control study by Buraczynska M et al. reported that the T allele of miR-196a2 rs11614913 could have protective effects for T2DM; conversely, the T allele and the TT genotype act as risk factors for subgroup patients with cardiovascular disease [52]. In a meta-analysis, including 1082 cases and 3102 controls, the rs11614913 T allele significantly decreased the T2DM risk [38]. Of note, the major allele in the Caucasian population is the C allele, whereas in the Chinese population, the major allele is T [53]. 

These discrepant findings may be due to the sample size included and to ethnic differences in genotype distributions. It is well known that the frequency of alleles of different miRNA SNPs are different between Caucasian and Asian populations. Also, sample size may affect results regarding the association between alleles and T2DM. On the other hand, the correlation between SNPs and the occurrence of T2DM could be affected by environmental risk factors and gene–environment interactions, with T2DM exhibiting genetic heterogeneity in different populations. 

In the present study, because we did not collect more data concerning clinical parameters in T2DM and control subjects, we could not perform further analysis of the association of these SNPs with other T2DM variables; this may constitute a limitation of our preliminary study, along with the small sample size of both groups. Additionally, subject selection bias cannot be ruled out. Finally, the genetic and biological effects of other miRNA variants or target genes should be further considered.

## 5. Conclusions

In conclusion, this preliminary study shows that miR-499a rs3746444 A>G may influence T2DM and DPN susceptibility, with carriers of the GG genotype and the G allele being at an increased risk in the Romanian population. Our results should be interpreted in a cautious and non-decisive manner, and additional observational studies on different and larger ethnic populations are essential.

## Figures and Tables

**Table 1 genes-14-01543-t001:** Subject characteristics.

Variable	T2DM	Control
N	167	324
Male/Female	86/81	174/150
Age (years), mean ± SD	66.12 ± 10.54	65.49 ± 7.94
BMI (kg/m^2^)	29.71	
Fasting plasma glucose (mg/dL)	172	
HbA1c (%)	7.2	
Diabetic polyneuropathy	89	

**Table 2 genes-14-01543-t002:** Minor allele frequencies (MAF) and Hardy–Weinberg equilibrium (HWE) values in the control group.

miRNA SNP	MAF	χ2	*p*
miR-27a rs895819 T>C	0.30	0.03	0.86
miR-146a rs2910164 G>C	0.26	0.25	0.61
miR-196a2 rs11614913 C>T	0.37	0.09	0.76
miR-499a rs3746444 A>G	0.23	1.51	0.22

**Table 3 genes-14-01543-t003:** Different inheritance model analyses of the miRNA SNPs between the T2DM and the controls.

Polymorphism	T2DM (n = 167)	Control (n = 324)	OR (95%CI)	*p* Value
**miR-27a rs895819**				
Codominant				
TT	77 (46.11%)	159 (49.07%)	Reference	-
TC	66 (39.52%)	135 (41.67%)	1.01 (0.67–1.51)	0.96
CC	24 (14.37%)	30 (9.26%)	1.65 (0.90–3.01)	0.105
Dominant				
TT	77 (46.11%)	159 (49.07%)	Reference	-
CC + TC	90 (53.89%)	147 (50.93%)	1.13 (0.77–1.64)	0.53
Recessive				
TT + TC	143 (85.63%)	294 (90.74%)	Reference	-
CC	24 (14.37%)	30 (9.26%)	1.64 (0.93–2.92)	0.092
Allelic				
T	220 (65.87%)	453 (69.91%)	Reference	-
C	114 (34.13%)	195 (30.09%)	1.20 (0.91–1.59)	0.19
**miR-146a rs2910164**				
Codominant				
GG	97 (58.09%)	181 (55.86%)	Reference	-
GC	64 (38.32%)	120 (37.04%)	0.99 (0.67–1.47)	0.98
CC	6 (3.59%)	23 (7.10%)	0.49 (0.19–1.24)	0.11
Dominant				
GG	97 (58.09%)	181 (55.86%)	Reference	-
CC + GC	70 (41.91%)	143 (44.14%)	0.91 (0.63–1.33)	0.64
Recessive				
GG + CG	161 (96.41%)	301 (92.90%)	Reference	-
CC	6 (3.59%)	23 (7.10%)	0.49 (0.19–1.22)	0.104
Allelic				
G	258 (77.25%)	482 (74.38%)	Reference	-
C	76 (22.75%)	166 (25.62%)	0.85 (0.63–1.17)	0.32
**miR-196a2 rs11614913**				
Codominant				
CC	79 (47.31%)	131 (40.43%)	Reference	-
CT	71 (42.51%)	148 (45.68%)	0.79 (0.53–1.18)	0.26
TT	17 (10.18%)	45 (13.89%)	0.63 (0.34–1.17)	0.13
Dominant				
CC	79 (47.31%)	131 (40.43%)	Reference	-
TT + CT	88 (52.69%)	193 (59.57%)	0.76 (0.52–1.10)	0.14
Recessive				
CC + CT	150 (89.82%)	279 (86.11%)	Reference	-
TT	17 (10.18%)	45 (13.89%)	0.70 (0.39–1.27)	0.23
Allelic				
C	229 (68.56%)	410 (63.27%)	Reference	-
T	105 (31.44%)	238 (36.73%)	0.79 (0.59–1.05)	0.098
**miR-499a rs3746444**				
Codominant				
AA	74 (44.31%)	189 (58.33%)	Reference	-
AG	76 (45.51%)	122 (37.66%)	1.59 (1.07–2.35)	0.02
GG	17 (10.18%)	13 (4.01%)	3.34 (1.54–7.28)	0.002
Dominant				
AA	74 (44.31%)	189 (58.33%)	Reference	-
GG + AG	93 (55.69%)	135 (41.67%)	1.76 (1.21–2.56)	0.003
Recessive				
AA + AG	150 (89.82%)	280 (95.99%)	Reference	-
GG	17 (10.18%)	13 (4.01%)	2.71 (1.28–5.73)	0.009
Allelic				
A	224 (67.07%)	500 (77.16%)	Reference	-
G	110 (32.93%)	148 (22.84%)	1.66 (1.24–2.22)	0.001

**Table 4 genes-14-01543-t004:** Association between miRNA SNPs and risk of DPN.

Polymorphism	With DPN(n = 89)	Without DPN(n = 78)	OR (95%CI)	*p* Value
**miR-27a rs895819**				
Codominant				
TT	43 (48.31%)	34 (43.59%)	Reference	-
TC	34 (38.20%)	32 (41.03%)	0.84 (0.43–1.63)	0.61
CC	12 (13.49%)	12 (15.38%)	0.79 (0.32–1.98)	0.62
C carriers	46 (51.69%)	44 (56.41%)	0.83 (0.45–1.52)	0.54
**miR-146a rs2910164**				
Codominant				
GG	55 (61.80%)	42 (53.85%)	Reference	-
GC	31 (34.83%)	33 (42.30%)	0.72 (0.38–1.35)	0.3
CC	3 (3.37%)	3 (3.85%)	0.76 (0.15–3.98)	0.75
C carriers	34 (38.20%)	36 (46.15%)	0.72 (0.39–1.34)	0.3
**miR-196a2 rs11614913**				
Codominant				
CC	48 (53.93%)	31 (39.74%)	Reference	-
CT	32 (35.96%)	39 (50.00%)	0.53 (0.28–1.01)	0.06
TT	9 (10.11%)	8 (10.26%)	0.73 (0.25–2.09)	0.55
T carriers	41 (46.07%)	47 (60.26%)	0.56 (0.30–1.04)	0.06
**miR-499a rs3746444**				
Codominant				
AA	31 (34.83%)	43 (55.13%)	Reference	-
AG	44 (49.44%)	32 (41.03%)	1.91 (0.99–3.65)	0.051
GG	14 (15.73%)	3 (3.85%)	6.47 (1.71–24.47)	0.002
G carriers	58 (65.17%)	35 (44.87%)	2.30 (1.23–4.29)	0.008

**Table 5 genes-14-01543-t005:** Association between miRNA SNPs and metabolic factors.

	Genotype			*p* Value
**miR-27a rs895819**	TT	TC	CC	
Frequency	77 (46.1%)	66 (39.5%)	24 (14.4%)	-
Age, years	66.51 ± 10.40	65.14 ± 10.80	67.63 ± 10.50	0.561
BMI (kg/m^2^)	29.92 ± 4.17	29.85 ± 4.53	28.67 ± 4.21	0.444
Glycemia	176.05 ± 70.80	166.52 ± 68.50	178.36 ± 69.00	0.652
HbA1c%	7.39 ± 2.05	6.98 ± 1.96	7.21 ± 2.20	0.505
**miR-146a rs2910164**	GG	GC	CC	
Frequency	97 (58.1%)	64 (38.3%)	6 (3.6%)	-
Age, years	66.69 ± 10.07	65.42 ± 11.44	64.50 ± 9.094	0.705
BMI (kg/m^2^)	30.19 ± 4.47	29.35 ± 3.99	25.85 ± 3.25	0.039
Glycemia	171.21 ± 70.77	176.21 ± 69.95	158.82 ± 40.01	0.772
HbA1c%	7.17 ± 2.08	7.29 ± 2.02	6.74 ± 1.32	0.816
**miR-196a2 rs11614913**	CC	CT	TT	
Frequency	79 (47.3%)	71 (42.5%)	17 (10.2%)	-
Age, years	66.90 ± 9.69	65.85 ± 11.30	63.71 ± 11.35	0.507
BMI (kg/m^2^)	29.54 ± 4.22	29.48 ± 4.28	31.49 ± 4.79	0.201
Glycemia	176.93 ± 73.10	168.63 ± 68.31	169.15 ± 57.75	0.75
HbA1c%	7.45 ± 2.16	7.01 ± 1.92	6.80 ± 1.72	0.307
**miR-499a rs3746444**	AA	AG	GG	
Frequency	74 (44.3%)	76 (45.5%)	17 (10.2%)	-
Age, years	65.31 ± 11.09	66.55 ± 9.74	67.76 ± 11.88	0.616
BMI (kg/m^2^)	30.13 ± 4.61	29.53 ± 4.00	28.69 ± 4.38	0.412
Glycemia	182.66 ± 75.98	160.54 ± 48.23	182.84 ± 80.42	0.121
HbA1c%	7.53 ± 2.25	6.82 ± 1.66	7.42 ± 2.30	0.102

## Data Availability

All the data presented here are available from the authors upon reasonable request.

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
