# Peer review of "miR-499a rs3746444 A>G Polymorphism Is Correlated with Type 2 Diabetes Mellitus and Diabetic Polyneuropathy in a Romanian Cohort: A Preliminary Study"

_genes, 2023, doi:10.3390/genes14081543_

Round 1

Reviewer 1 Report

Emilia Burada et al. examined the correlation between miRNA SNP and T2DM susceptibility and found miR-499a rs3746444 A>G SNP significantly correlated with T2DM in a Romanian cohort. I have some comments.

It’s better to include the justification of sample size determination and the statistical power.

Why did the authors select these miRNA SNP for the study?

Why are the demographic and Clinical Characteristics for control incomplete?

Have the authors considered the adjustment by age, sex, clinical symptoms, etc in logistic regression analysis?

In the discussion, the authors could try to cite papers to explain how SNP affects target expression or could change the T2DM traits. add more literature related to the role of potential targets of miRNAs in T2DM.

Reviewer 2 Report

The manuscript is well written and scientifically sound. The conclusion made by the authors is well supported by the data. However, it is unclear what made the authors choose these four miRNAs SNP (miR-27a rs895819, miR-146a rs2910164, 31 miR-196a2 rs11614913 and miR-499 rs3746444) to correlate among diabetes patients in the Romanian population. The authors should give the explanation/justification for focussing on these four miRNAs SNPs on which that entire study is based. 
